# The effects of psychological flexibility and night shifts on mental health and well-being in nurses

Xinhong Li[1], Juan Han[2], Hongmei Lin[3]*

1 Department of Outpatient Surgery, Central Hospital Affiliated to Shandong First Medical University, Jinan, Shandong Province, China, 2 Hyperbaric Oxygen, Jinan Central Hospital, Jinan, Shandong Province, China, 3 Department of Gynecology, Central Hospital Affiliated to Shandong First Medical University, Jinan, Shandong Province, China

* meihonglin21@gmail.com

**Data Availability Statement:** All relevant data are within the manuscript.

**Funding:** This study was funded by National Natural Science Foundation of China (grant number: 81701159), Taishan Scholar Project of

## Abstract

### Background

Examining mental health among nurses in the later stages of the COVID-19 pandemic could offer valuable information for addressing these symptoms in the long term. Therefore, the current study aimed to assess the mental health and well-being of Chinese nurses, as well as investigate the impact of psychological flexibility and night shifts on this relationship.

### Method

In cross-sectional, hospital-based, multicenter study, 422 Chinese nurses were selected by multistage stratified cluster random sampling. The mental health status, psychological flexibility, and wellbeing were assessed via 12-item General Health Questionnaire (GHQ-12), Chinese version of Personalized Psychological Flexibility Index, and Chinese version of the 5-item WHO Well-Being Index, respectively. To examine the proposed theoretical model, we conducted structural equation modeling using SPSS Amos 26 version. The age, gender, night shift, psychological flexibility, mental health, and well-being data were entered into the model. Pearson correlation and chi-square were used to explore the correlation between variables.

### Results

The high night shifts, being young and low psychological flexibility significantly had a direct effect on worse mental health (night shifts: ES, 95% CI: 0.619, 0.328–0.725; age: ES, 95% CI: 0.542, 0.226–0.993; psychological flexibility: ES, 95% CI: 0.675, 0.369–1.466). The low psychological flexibility and worse mental health were able to directly effect on worse wellbeing (psychological flexibility: ES, 95% CI: 0.419, 0.757–1.519; mental health: ES, 95% CI: 0.719, 1.109–2.607). In addition, psychological flexibility through the mediation of mental health also had an indirect effect on wellbeing (ES, 95% CI: 0.269, 0.957–2.165).

Shandong Province of China (grant number: 202103200), and Youth Scientific Research Fund of Liaocheng People's Hospital (grant number: 201910915). "The funders had no role in study design, data collection and analysis, decision to publish, or preparation of the manuscript.

**Competing interests:** The authors have declared that no competing interests exist.

## Conclusions

Being young, having more night shifts and having less psychological flexibility can be related to the deterioration of mental health and well-being in nurses. Therefore, it is recommended that nurses use the shift routine program with the least focus on the night shifts. Also, interventions to teach younger nurses how to face work stress and interventions to improve the psychological flexibility of all nurses are needed.

## Introduction

Mental health issues among healthcare workers have been identified as a significant factor impacting the quality of healthcare services [1]. Healthcare workers globally are experiencing various mental health challenges such as depression, stress, anxiety, and burnout [2]. The prevalence of these problems has been reported in some studies. For example, some studies showed that burnout is high among Jordanian nurses [3, 4]. Also, Anwar et al. (2022) reported that 64.15% and 33.19% of nurses in Faisalabad suffer from moderate depression and severe anxiety, respectively [5]. Some demographic factors are related to burnout in nurses. For example, some studies have shown that the symptoms of depersonalization are more in male nurses than in female nurses, and signs of emotional exhaustion are more in female nurses than in male nurses [6]. Some studies have been reported that single Taiwanese and Nigerian nurses suffer more burnout [7, 8], while other studies in Lebanon and Rwanda reported opposite results [9, 10].

In China, the prevalence of burnout and mental health disorders among healthcare workers, including nurses, is on the rise [11]. Some studies have indicated that the overall prevalence of burnout symptoms among nurses worldwide is approximately 11.23% [12]. Additionally, a study focusing on Chinese tertiary public hospital nurses revealed that the prevalence rates of depression, anxiety, and stress were 17.9%, 34.4%, and 12.7%, respectively [13]. In nursing, which is considered one of the most high-stress occupations, mental health is not only relates to the well-being of nurses, but also has implications for their job satisfaction, career stability, and intention of leaving the occupation [14]. Furthermore, it is important to note that poor mental health among nurses can have a negative impact influence on the quality of healthcare services provided and can lead to increased healthcare expenditures.

It is crucial to pinpoint the factors that impact mental health of nurses to support these health services providers in their struggle against the psychological problems. Psychological flexibility is an individual difference factor that can serve as a valuable resource for nurses [15]. Some studies have indicated that adopting a stance objective and accepting attitude towards one's feelings and thoughts, acting in accordance with personal values, and remaining aware of present moment experiences can lead to positive mental health [16]. Also, Kılıç et al. (2022) has shown that self-report measures of psychological flexibility was able to predict mental health over the span of one or more years [17]. Additionally, Hernández-López et al. (2021) and Landi et al. (2022) have indicated that psychological flexibility can also influence distress levels in the short term (periods of two months and nine months during the first year of the COVID-19 pandemic) [18, 19].

Heavy workloads, especially when night shifts are common, are a major cause of increased mental health problems. Numerous studies have consistently found that working night shifts is associated with a higher risk of experiencing depression, anxiety, burnout, sleep problems, and a lower overall quality of life among healthcare workers such as nurses [20, 21]. Nurses

need to stay alert and attentive to provide optimal care to patients. Working night shifts can leave nurses tired and lacking sleep, putting patients at risk [22]. Some studies have shown that the longer someone works as a shift healthcare worker, the more likely they are to experience burnout [23]. Also, in meta-analysis study, it was seen that female nurses who work in shifts (especially night shift) are more prone to depression than male nurses [24]. Given that most nurses are women, this is a significant issue.

In addition, night shift in nurses can be one of the risk factors of some diseases including cancer. Some studies have found that night shifts for several consecutive years can significantly influence the risk of breast cancer in nurses [25–27]. Also, a review study reported that there was a relationship between prolonged rotating night shifts and breast cancer in nurses [28].

Mental health and well-being among nurses is a challenging issue. Identifying the different dimensions of this challenge can help in evaluating the problem. One of the important issues that can be evaluated in this challenge is the night shift and some mental capabilities of nurses, including psychological flexibility. Considering that night shift is mandatory among many nurses, especially nurses working in hospitals, evaluating its relationship with mental health and well-being can provide better insight into this issue. Therefore, the current study aimed to assess the mental health and well-being of Chinese nurses, as well as investigate the impact of psychological flexibility and night shifts on this relationship.

## Methods and materials

### Study design and participants

We conducted a cross-sectional, hospital-based, multicenter study in Shanghai, China, from September 20 to December 20, 2023. Chinese public hospitals are divided into three levels based on the level of quality: Grade III, Grade II, and Grade I. We only selected Grade II hospitals to ensure equal conditions. In addition, to equalize the conditions, only the nurses of the hospitalization (inpatient) section were evaluated.

Using a stratified cluster sampling method, 10 Grade II public hospitals in Shanghai were selected. The sample size was calculated by using the PS software (Power and Sample Size Program, version 3.1.2). Based on previous studies [29] and considering $\alpha = 0.05$ and power = 80%, the number obtained was equal to 368 participants. Finally, due to the possibility of losing 5% of the samples, the final required sample size was 387 participants.

The inclusion criteria were as follows: nurses working in hospitalization (inpatient) section of Grade II hospitals, nurses aged 18–60 years old, nurses with at least one year of working experience, and nurses willing to provide consent for participation in the study. On the other hand, the following nurses were excluded based on the criteria below: nurses who had taken a leave for more than six months in the previous year due to different reasons, nurses who were absent from work (either on leave or vacation) during the administration of the questionnaire, and nurses who were not directly involved in patient care, such as the head of the nursing department and head nurse. It's important to note that all participants in the study provided signed informed consent. The written questionnaires were distributed among the participants with help of head nurses. After filling the questionnaires, the data were entered into Excel software (2016 version). To maintain the confidentiality of participants' information, there was no information about the name and identity of the participants in the questionnaire.

Values were assigned to various variables such as age, gender, night shifts number in the last month, mental health, well-being, and psychological flexibility for modeling purposes, as outlined in Table 1.

**Table 1. Assignment of observed variables.**

| Variables | Value |
|---|---|
| 1. Age | 18–64, continuous variable |
| 2. Gender | 1 = Male, 2 = Female |
| 3. Night shift | ≥0, continuous variable |
| 4. Psychological flexibility | 1–7, continuous variable |
| 5. Mental health score | 0–7, continuous variable |
| 6. Well-being | 0–5, continuous variable |

The research ethics committee of Central Hospital Affiliated to Shandong First Medical University (YL20231051X) has approved this manuscript. Also, written consent was received from the participants and the research was conducted based on the Helsinki Declaration.

## Mental health

The GHQ-12 is used to assess the mental health of nurses. It consists of 12 self-assessment items, each with four options (A, B, C, and D). The scoring method is bimodal, with scores of 0-0-1-1. Here's how the scoring works: selecting A or B: Score of 0 and choosing C or D: Score of 1. A cumulative score of ≥ 4 indicates a positive mental health screening result, suggesting conditions like anxiety, depression, or insomnia [30]. A higher score reflects a more severe mental health condition.

The Chinese versions of GHQ-12 for different professional groups, healthcare workers, and the general population have shown strong internal consistency, reliability, and validity [31, 32]. In this study, the scale had a Cronbach's α coefficient of 0.93.

## Psychological flexibility

The PPFI (Psychological Flexibility Inventory) was used to measure psychological flexibility. This inventory consists of three aspects: acceptance, avoidance, and harnessing, with a total of 15 statements. Participants were asked to rate their agreement level on a 7-point Likert scale, ranging from 1 (completely disagree) to 7 (fully agree). The avoidance subscale was scored in reverse. The Chinese version of this questionnaire has already been approved [33, 34]. In this study, the scale had a Cronbach's α coefficient of 0.87.

## Well-being

WHO-5 is a questionnaire that assesses wellbeing using five statements. Participants rate each statement on a scale of 0 (never) to 5 (all the time). A higher score indicates a higher level of wellbeing. The Chinese version of this questionnaire has already been approved [35, 36]. In this study, the scale had a Cronbach's α coefficient of 0.85.

## Statistical analysis

We summarized demographic data, mental health scores, nursing care satisfaction scores, and relational care scores using measures such as percentages, means, and standard deviations. Also, the relationship between different variables was examined using a correlation matrix. For statistical analysis, we used SPSS version 26.0 (IBM Corp., Armonk, NY, USA). Proportions were utilized for qualitative and ordinal data, while means and standard deviations (SD) were used for quantitative data. To examine the proposed theoretical model, we conducted structural equation modeling using SPSS Amos 26 version. The fit of the model was assessed

using various fitness indices including goodness-of-fit index (GFI), normal fit index (NFI), comparative fit index (CFI), adjusted goodness-of-fit index (AGFI), root mean square error of approximation (RMSEA), and tucker-lewis index (TLI). GFI >0.95 [37], NFI >0.95 [38], CFI >0.96 [39], AGFI >0.90 [37], RMSEA <0.05 [37], and TLI >0.90 [37] indicated the fit of the model.

## Results

In general, 612 questionnaires were distributed among nurses. 564 questionnaires were filled by the participants (the response rate was equal to 92.15%). However, only 422 questionnaires were approved (the correct response rate was equal to 74.82%) (27.34 ±4.19 years) and the rest of the questionnaires were not completely filled and therefore were excluded from the study.

64% of nurses were women. The mean of night shift per last month, weekly hours of nurses, and working years were 4.822 ±4.542 night, 41.185 ±4.314 hours and 7.00±3.815 years, respectively. Also, the mean score of mental health, psychological flexibility, and well-being were 2.607±2.098, 4.571±1.936, and 3.342±1.532. Also, the positive rate of mental health status was 22.91% (About 22.91% had mental problems).

Pearson correlation and chi-square, and nonparametric tests were used to explore the correlation between variables.

Age was negatively associated with night shift ($r = -0.226$) and mental health scale score ($r = -0.142$). In contrast, there was a positive correlation between age and psychological flexibility ($r = 0.112$). Female tend to have a stronger connection with psychological flexibility compared to male ($\chi^2 = 4.599$). Night shift had a negative correlation with psychological flexibility (r = -0.166) and a positive correlation with mental health scale score ($r = 0.277$). Psychological flexibility was negatively correlated with mental health scale score ($r = -0.226$), and positively correlated with well-being ($r = 0.221$). There was also a negative correlation between mental health scale score and well-being ($r = -0.315$) (Tables 2 and 3).

The age, gender, night shift, psychological flexibility, mental health, and well-being data were entered into the model. The final structural equation model fitted well with the study data (CFI = 0.992, NFI = 0.959, TLI = 0.901, GFI = 0.984, AGFI = 0.929, RMSEA = 0.049).

**Table 2. Correlation between study variables.**

|  | Age | Night shift | Psychological flexibility | Mental health | Well-being |
|---|---|---|---|---|---|
| Age | 1 | -0.226* | 0.112* | -0.142* | 0.081 |
| Night shift | -0.226* | 1 | -0.166* | 0.277* | 0.085 |
| Psychological flexibility | 0.112* | -0.166* | 1 | -0.226* | 0.221* |
| Mental health | -0.142* | 0.277* | -0.226* | 1 | -0.315* |
| Well-being | 0.081 | 0.085 | 0.221* | -0.315* | 1 |

*$P<0.05$

**Table 3. Chi-square test and nonparametric test results for study variables ($\chi^2$ / Z value).**

|  | Age | Gender | Night shift | Psychological flexibility | Mental health | Well-being |
|---|---|---|---|---|---|---|
| Gender | 0.073 | 1 | -0.049 | 4.599* | 2.261 | 0.223 |

*$P<0.05$

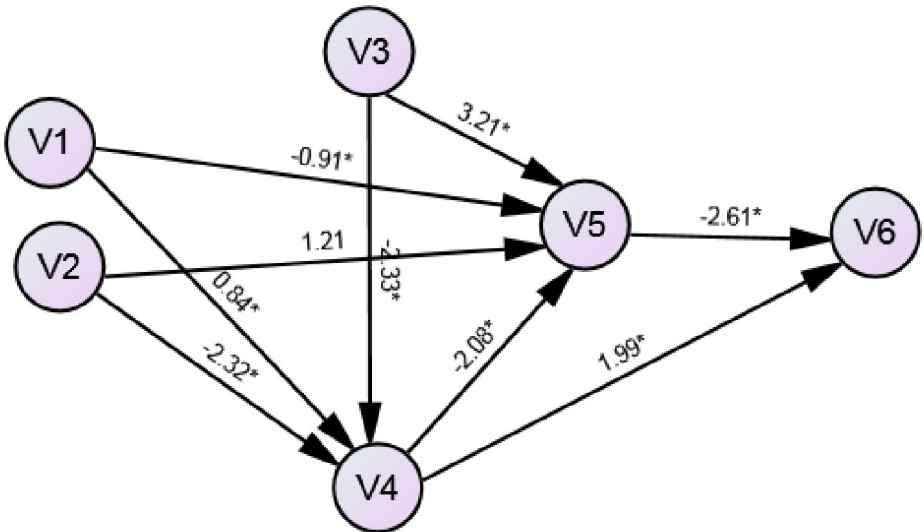

**Fig 1. Modified model; \*the goodness-of-fit indices were: CFI = 0.992, NFI = 0.959, TLI = 0.901, GFI = 0.984, AGFI = 0.929, RMSEA = 0.049.** \*$P<0.05$; V1 = age; V2 = gender; V3 = night shift; V4 = psychological flexibility; V5 = mental health; V6 = well-being.

\*$p<0.05$) (Fig 1). Except for the effect of gender on the mental health and well-being were not significant; other effects were all significant ($p<0.05$).

Table 4 shows the standardized direct, indirect, and total effects, and effect size (ES) the 95% confidence interval (CI) for each construct. Eight direct, three indirect, and ten total effects were significant.

The relationship between night shifts and age with mental health and psychological flexibility showed that the high night shifts and being young significantly had a direct effect on worse mental health (night shifts: ES, 95% CI: 0.619, 0.328–0.725; age: ES, 95% CI: 0.542, 0.226–0.993) and low psychological flexibility (night shifts: ES, 95% CI: 0.199, 0.145–0.421; age: ES, 95% CI: 0.130, 0.140–0.609). Also, these variables were able to influence worse well-being

**Table 4. Standardized direct, indirect, and total effects of variables in the final model (ES, 95%CI).**

| Exogenous variable | Endogenous variable | Direct effect | Indirect effect | Total effect |
|---|---|---|---|---|
| Age | mental health | 0.542, 0.226–0.993\* | | 0.542, 0.226–0.993\* |
| | psychological flexibility | 0.130, 0.140–0.609\* | | 0.130, 0.140–0.609\* |
| | well-being | | 0.151, 0.226–0.852\* | 0.151, 0.226–0.852\* |
| Gender | mental health | 0.219, 0.282–0.431 | | 0.219, 0.282–0.431 |
| | psychological flexibility | 0.062, 0.026–0.082\* | | 0.062, 0.026–0.082\* |
| | well-being | | 0.229, 0.055–0.179 | 0.229, 0.055–0.179 |
| Night shift | mental health | 0.619, 0.328–0.725\* | | 0.619, 0.328–0.725\* |
| | psychological flexibility | 0.199, 0.145–0.421\* | | 0.199, 0.145–0.421\* |
| | well-being | | 0.367, 0.871–1.959\* | 0.367, 0.871–1.959\* |
| Psychological flexibility | mental health | 0.657, 0.369–1.466\* | | 0.657, 0.369–1.466\* |
| | well-being | 0.419, 0.757–1.519\* | 0.269, 0.957–2.165\* | 0.688, 1.714–3.684\* |
| Mental health | well-being | 0.719, 1.109–2.607\* | | 0.719, 1.109–2.607\* |

\* $p<0.05$

indirectly and through the mediation of mental health and psychological flexibility (night shifts: ES, 95% CI: 0.367, 0.871–1.959; age: ES, 95% CI: 0.151, 0.226–0.852). Gender (being female) was only able to directly predict high psychological flexibility (ES, 95% CI: 0.062, 0.026–0.082). The low psychological flexibility was able to directly effect on worse mental health (ES, 95% CI: 0.675, 0.369–1.466) and wellbeing (ES, 95% CI: 0.419, 0.757–1.519). In addition, psychological flexibility through the mediation of mental health also had an indirect effect on wellbeing (ES, 95% CI: 0.269, 0.957–2.165). The worse mental health was able to directly effect on worse wellbeing (ES, 95% CI: 0.719, 1.109–2.607). In the model, the strongest effects were the effect of mental health on well-being (ES, 95% CI: 0.719, 1.109–2.607), and effect of psychological flexibility on well-being (ES, 95% CI: 0.688, 1.714–3.684).

## Discussion

The purpose of this study was to explore whether the association between mental health and well-being in Chinese nurses can be explained by other factors such as night shift and psychological flexibility. As expected, there was a correlation between worse mental health status and worse well-being in nurses, and mental health had a direct effect on well-being. These results are consistent with other studies results [40–42].

Based on the results of the present study, some factors had an effect on this relationship. For example, psychological flexibility could directly predict the improvement of well-being in nurses. Guerrini Usubini et al, (2021) showed that psychological flexibility played a significant role in explaining psychological well-being in adults with obesity [43]. Greville-Harris et al., (2024) reported the relationship between psychological flexibility and mental well-being in UK surgeons [44]. In another study, the relationship between psychological flexibility and emotional well-being was reported in Australian psychology students [45]. It was also shown in a meta-analysis that there was a link between psychological inflexibility and worse well-being [46]. Also, some studies have shown that psychological flexibility strongly correlates with various positive psychological outcomes, including reduced stress, anxiety, and depressive symptoms [47–49]. Psychological flexibility refers to the ability to stay connected with the present moment while pursuing long-term objectives instead of giving in to immediate desires. It enables individuals to adjust to shifting circumstances and respond with innovative, constructive, and values-driven behaviors [50]. By adapting to stressors and having flexible reactions, the feeling of overwhelmed or consumed by oneself occurs less often. This can lead to better stress management and improved overall well-being [51]. Furthermore, psychological flexibility is crucial for maintaining overall well-being [52].

In the current study, in addition to the direct relationship between psychological flexibility and will-being, there was also an indirect relationship between these two variables, which was mediated by mental health.

The association between psychological flexibility and mental health has been reported in other studies. For example, Prudenzi et al., (2023) reported an association between higher levels psychological flexibility with better mental health in British adults during the first wave of the COVID-19 pandemic [53]. Also, Chong et al., (2023) found that psychological flexibility might be a key factor that helps Hong Kong and Swiss nurses stay mentally strong and healthy, especially when dealing with burnout and stress during the COVID-19 pandemic [54].

It has been found that psychological flexibility can reduce stress, anxiety, depression, and other mental health issues. Additionally, being more psychologically flexible can have positive effects such as boosting confidence, improving self-esteem, and enhancing relationships. By cultivating psychological flexibility, we can develop a healthier mindset that allows us to navigate life's challenges with greater ease and resilience [45, 55]. In the present study, some factors

such as gender and age had an effect on psychologically flexibility. Based on the results of the present study, there was a relationship between being a woman and high psychological flexibility. These results were consistent with the study results of Fathima et al. (2022) [56]. One possible reason for this relationship is that females tend to be more open and expressive about their problems compared to men. This idea is supported by a meta-analysis review which discovered that women often use coping mechanisms like talking about their feelings, which helps them communicate their worries to others [57]. Additionally, due to the prevailing gender stereotypes in China, females are traditionally expected to be more adaptable and flexible in various aspects of life. This mindset is instilled in them from a young age, which could contribute to their higher levels of psychological flexibility.

In a study, Wang et al., (2022) reported that cognitive flexibility is higher in men and emotional flexibility is higher in women [57]. The difference between the present study and this study was that psychological flexibility was evaluated in general and its dimensions were not investigated separately.

Also, the current study showed that age had a direct effect on psychological flexibility. This means that older nurses (compared to younger nurses) have more psychological flexibility. These results are consistent with the results of Plys et al, (2022) [58] and inconsistent with the Zukerman et al, (2023) [59]. It seems that the different measurement tools and participants (nurses vs. non-nurses) can be among the reasons for this disparity. The previous studies have shown that the ability to use self- and emotion-regulation strategies, such as being mindful and accepting, improves with age [60, 61]. Therefore, it can be expected that psychological flexibility in nurses increases with age.

In general, it can be said that psychological flexibility was one of the factors affecting the mental health and well-being of nurses, which was lower in male nurses and younger nurses. Studies show that one of the ways to increase psychological flexibility is the use of Acceptance and Commitment Therapy (ACT) interventions [62, 63]. ACT focuses on pursuing important life areas and goals, like close relationships, fulfilling work, and personal development, even when facing difficult experiences. In ACT, this is achieved by developing psychological flexibility, which helps people stay involved in meaningful activities, even when they have negative thoughts, feelings, or face other challenges [63].

Another result of the current study was that night shift had an effect on well-being indirectly through influencing mental health and psychological flexibility. The relationship between night shift and mental health has been reported in other studies [64, 65]. However, we did not find any study that evaluated the relationship between night shifts and psychological flexibility in nurses. Working the night shift can have negative effects on brain. A 2019 study discovered that individuals who work night shifts for an extended period experience the following impaired working memory, processing speed, and cognitive flexibility after a night shift and more operational errors, occupational accidents, and injuries [66]. These issues are believed to arise from the disruption of the body's natural sleep patterns, also known as circadian rhythms, caused by night shift work. However, to keep the healthcare system running smoothly, many nurses and other healthcare workers, as well as non-healthcare staff, work at night, either on a rotating schedule or permanently. For example, nurses work at night to provide continuous care and monitor patients, as well as to give them medication. Lack of sleep can lead to various difficulties such as trouble focusing and concentrating, and difficulty making decisions and solving problems. In addition, night shifts can lead to others various health issues such as persistent fatigue, behavioral changes like irritability, bad attitude, reduced communication skills, and reduced ability to cope with emotional demands at work, and reduced efficiency [67]. Some studies have shown that nurses who switched from night shifts to day shifts experienced a noticeable reduction in anxiety and depression symptoms over a two-year

period [68, 69]. Poor performance caused by lack of sleep can make employees feel like they've lost their sense of purpose, which can harm their mental health and well-being [68, 69]. It recommended that reducing the amount of work for night shift nurses, addressing the sources of stress during night shifts, and encouraging rest and relaxation could help with mental health. Nursing managers should give nurses more days off after night shifts, space out night shifts more, and reduce overtime to lower the number of sleep problems caused by shift work among nurses.

Despite the fact that the negative effects of night shift on mental and physical health are known, many nurses are forced to work night shift. Some studies have shown that some strategies can reduce the negative effects of night shift. For example, spending more time in light during the day, especially before a shift, can lessen the harmful impact of evening and night-time light on circadian phase delays, sleep, and melatonin levels [70]. Some dietary strategies can also reduce the negative effects of night shift. The Centers for Disease Control and Prevention (CDC) suggests some ways to handle work hours, like planning meals, eating high-protein foods during work, and not having big meals before bed. These tips can help people with shift work eat better, keep their energy up, and sleep well [71]. Also, regular exercise can help prevent and recover from issues by assisting shift workers in developing a more stable circadian phase, which improves their sleep and wake patterns [72].

Another variable that had an effect on the relationship between mental health and well-being of nurses in this study was age. Age had a direct effect on mental health and indirectly on well-being (through influencing mental health). The relationship between increasing age and improving mental health has been reported in some studies [73]. However and inconsistent with the results of the present study, Haghi et al, (2022) have shown that older nurses were more likely to be psychologically vulnerable and need more support [74]. A possible reason is that Haghi et al.'s study was conducted in a pandemic situation, while the present study was conducted in the post-pandemic era. The pandemic itself is a situation that causes the deterioration of mental health, especially in nurses who are in direct contact with this disease [75]. In contrast, Oates et al. (2017) reported that the number of years in a post was inversely correlated with the overall experience of mental health problems in nurses [76]. As nurses get older, they tend to become more mature and gain more work experience. This increase in maturity and experience is likely the reason why older nurses perceive less stress and mental health problems compared to their younger counterparts [77].

It is important to acknowledge some limitations of this study. The use of a cross-sectional design prevents the identification of a causal relationship between variables. It is suggested that longitudinal studies be conducted to evaluate the relationship between the variables of this study. One of the limitations of the present study was that it only conducted in hospitalization section of Grade II public hospitals. Therefore, caution should be taken in generalizing its results to other sections and hospitals (private hospitals and public hospitals of different levels). The present study was conducted only in Shanghai city. Due to the economic and cultural differences of different Chinese cities, caution should be taken in generalizing its results to other cities and regions of China. Although valid tools were used to measure data, it was collected by self-report. Therefore, caution should be taken in interpreting the results.

## Conclusion

This study clearly shows a significant relationship between night shifts, age, and psychological flexibility with mental health and well-being in Chinese nurses. This indicates that nurses who work more night shifts have worse mental health and well-being. It seems that the possible effects of night shift on mental health and well-being can be moderated by proper shift

planning, workplace standardization and paying attention to the issue of sleep deprivation that occurs during night shifts. This study showed that younger nurses have worse mental health and well-being. By using appropriate interventions, this target group can be educated more so that they gradually learn how to deal with job stressors. It is also possible to consider fewer night shifts for nurses in the early years of their career. The low psychological flexibility and worsening mental health and well-being were also reported in this study. It seems that psychological flexibility can be evaluated during the recruitment of nurses, so that nurses with higher flexibility can be hired. Also, interventions can be used to promote psychological flexibility during working years.

## Acknowledgments

We thank all the nurses who participated in this study.

## Author Contributions

**Conceptualization:** Xinhong Li, Juan Han, Hongmei Lin.

**Data curation:** Xinhong Li, Juan Han, Hongmei Lin.

**Formal analysis:** Xinhong Li, Hongmei Lin.

**Investigation:** Xinhong Li, Juan Han, Hongmei Lin.

**Methodology:** Juan Han, Hongmei Lin.

**Project administration:** Juan Han, Hongmei Lin.

**Resources:** Hongmei Lin.

**Software:** Hongmei Lin.

**Supervision:** Xinhong Li, Juan Han, Hongmei Lin.

**Validation:** Xinhong Li, Hongmei Lin.

**Visualization:** Hongmei Lin.

**Writing – original draft:** Xinhong Li, Juan Han, Hongmei Lin.

**Writing – review & editing:** Hongmei Lin.

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
