## [Decision Letter · Decision Letter 0]

3 Oct 2024

PONE-D-24-36000The effects of psychological flexibility and night shifts on the relationship between mental health and well-being in Chinese nursesPLOS ONE

Dear Dr. Lin,

Thank you for submitting your manuscript to PLOS ONE. After careful consideration, we feel that it has merit but does not fully meet PLOS ONE’s publication criteria as it currently stands. Therefore, we invite you to submit a revised version of the manuscript that addresses the points raised during the review process.

**THERE ARE 2 REVIEWERS TO RESPOND, AND ALSO A NUMBER OF COMMENTS BY THE ACADEMIC EDITOR.**

We look forward to receiving your revised manuscript.

Kind regards,

Javier Fagundo-Rivera, PhD

Academic Editor

PLOS ONE

**Journal Requirements:**

**Additional Editor Comments:**

**Dear authors,**

**Two reviewers have send their comments, and also you have a number of comments by the Editor to respond.**

**EDITOR COMMENTS:**

**1. Abstract. The Results section is a bit reiterative. The same information is given with and without numerical data. Try to sum up the information.**

**2. Abstract. Conclusions are very short and should give more information on night shifts.**

**3. Introduction. Increase the literature evidence about night shifts. The information about night shifts and its health-related implications should be enhaced. Also, the IARC puts night shifts as predictor of cancer. Night shifts have implications on work-family conflicts and changes in lifestyle too. Examples can be found here: https://pubmed.ncbi.nlm.nih.gov/33321692/ ; https://pubmed.ncbi.nlm.nih.gov/33806956/ ; https://pubmed.ncbi.nlm.nih.gov/34070908/.**

**4. Methods. Increase the information about the study procedure. Did all nurses worked in the same clinical areas or there were differences (emergency, surgery, hospitalization, infectious diseases...)? Was it a written or virtual-based survey? If written, how did you computerized the responses? What program did you use to compile information before applying AMOS? Include information about anonymization of the responses.**

**5. Results. Information resulting from Table 4 is confusing (difficult to read), I recommend to check the paragraph about direct and indirect effects.**

**6. Discussion. Information about night shifts is general, more appropriate for the Introduction. Whereas, authors should increase relations between night shifts and mental health, psychological well-being.**

**7. Limitations. You mention the number of questions, how many questions were in the questionnaire? how much time did it take to complete the survey? Did participants refused to complete the survey because they were tired? I do not see information about nurses areas of work (emergency, surgery, hospitalization...), so there might be important differences depending on the area that the authors did not consider.**

**8. Conclusions. The conclusions are vague, and some numerical data and principal variables should be mentioned.**

**Reviewers' comments:**

Reviewer's Responses to Questions

**Comments to the Author**

1. Is the manuscript technically sound, and do the data support the conclusions?

Reviewer #1: Yes

Reviewer #2: Yes

2. Has the statistical analysis been performed appropriately and rigorously? 

Reviewer #1: Yes

Reviewer #2: Yes

3. Have the authors made all data underlying the findings in their manuscript fully available?

Reviewer #1: Yes

Reviewer #2: Yes

4. Is the manuscript presented in an intelligible fashion and written in standard English?

Reviewer #1: Yes

Reviewer #2: Yes

5. Review Comments to the Author

**Reviewer #1: **

Dear authors,

I have read your paper with great interest. The paper is rich in information that provides valuable information in the field of nursing. This study aims to assess the mental health and well-being of Chinese nurses, as well as investigate the impact of psychological flexibility and night shifts on this relationship. I believe implementing the following suggestions will enhance the overall quality of your paper:

Title:

Please consider revising the title to make no more than 15 words.

Abstract:

the 10 Grade II public hospitals. I think this information is not necessary. Please add the statistical issues in your abstract.

Conclusion:

social support in the workplace helps young nurses cope better with job stress and protect their mental health and wellbeing. How you conclude that since your study does not investigate social support. Please re-write your conclusion to be consistent with your findings.

Keywords: add flexibility word.

Introduction:

The introduction is well-written. However, there are some missing paragraphs for demographic factors included in your study. Why they are important and what the literature says about them. I highly recommend before the method section to write the significant values of your study and make a good connection between your variables.

Suggestion references to read.

Burnout among nurses and teachers in Jordan.

Self-evaluation and professional status as predictors of burnout among nurses.

Method:

-public hospitals: how many?

-Clarify the data time collection period.

-Address the issue of data collection such as how many nurses respond and what is the response rate. (Maybe as an appendix Table).

Add a reference for the normal values of CFI, TLI, REMSA…etc in your analysis section.

Results

Appropriate.

Discussion:

-Please highlight your findings instead of repeating your aims.

-being more psychologically flexible. Please how nurses can be??? Write more details in application.

-write more details regarding the connection between your variables.

- I noticed that the majority of references were from China. I respect that. But I suggest strengthening your paper by adding more international studies regarding your topic.

Finally, I hope to add clinical implications for nurses to reduce the effect of the night shift on their lives. Some nurses are obligated to work in night shift for several reasons (study, home). Please take this into your mind.

References:

Please double-check according to the journal style.

Tables:

Ensure that abbreviations used in tables are mentioned in the footnotes.

Best of luck with your revisions.

Kind regards,

**Reviewer #2: **

Thank you for reviewing of this paper. The manuscript is well written in standard English. The statistical Analysis method should be somewhat clear. The recommendation should be revised and rewritten based on your findings.

6. PLOS authors have the option to publish the peer review history of their article (what does this mean?). If published, this will include your full peer review and any attached files.

Reviewer #1: No

Reviewer #2: **Yes: **Worku Chekol Tassew

---

## [Author Response · Author response to Decision Letter 0]

8 Oct 2024

Response to comments: 

Editor comments: 

1. Abstract. The Results section is a bit reiterative. The same information is given with and without numerical data. Try to sum up the information.

Response: I edited: lines 40-46. 

2. Abstract. Conclusions are very short and should give more information on night shifts.

Response: I edited: lines 47-51. 

3. Introduction. Increase the literature evidence about night shifts. The information about night shifts and its health-related implications should be enhaced. Also, the IARC puts night shifts as predictor of cancer. Night shifts have implications on work-family conflicts and changes in lifestyle too. Examples can be found here: https://pubmed.ncbi.nlm.nih.gov/33321692/ ; https://pubmed.ncbi.nlm.nih.gov/33806956/ ; https://pubmed.ncbi.nlm.nih.gov/34070908/.

Response: I added: lines 105-109

4. Methods. Increase the information about the study procedure. Did all nurses worked in the same clinical areas or there were differences (emergency, surgery, hospitalization, infectious diseases...)? Was it a written or virtual-based survey? If written, how did you computerized the responses? What program did you use to compile information before applying AMOS? Include information about anonymization of the responses.

Response: I added: lines 121-122, 128, 136-139, 

5. Results. Information resulting from Table 4 is confusing (difficult to read), I recommend to check the paragraph about direct and indirect effects.

Response: I edited: lines 190-197, 215-230. 

6. Discussion. Information about night shifts is general, more appropriate for the Introduction. Whereas, authors should increase relations between night shifts and mental health, psychological well-being.

Response: I added: lines 287-290, 295-303. 

7. Limitations. You mention the number of questions, how many questions were in the questionnaire? how much time did it take to complete the survey? Did participants refused to complete the survey because they were tired? I do not see information about nurses areas of work (emergency, surgery, hospitalization...), so there might be important differences depending on the area that the authors did not consider.

Response: I removed this limitation, because I had not measured it. I added new limitations. Lines: 309-328. 

8. Conclusions. The conclusions are vague, and some numerical data and principal variables should be mentioned.

Response: I edited: lines: 330-342. 

Reviewer #1: 

Dear authors,

I have read your paper with great interest. The paper is rich in information that provides valuable information in the field of nursing. This study aims to assess the mental health and well-being of Chinese nurses, as well as investigate the impact of psychological flexibility and night shifts on this relationship. I believe implementing the following suggestions will enhance the overall quality of your paper:

Title:

Please consider revising the title to make no more than 15 words.

Response: The journal has not set a limit on the number of tile words. However, with the approval of the editor, I can change this title to: ‘’The effects of psychological flexibility and night shifts on mental health and well-being in nurses’’

Abstract:

the 10 Grade II public hospitals. I think this information is not necessary. Please add the statistical issues in your abstract.

Response: The place of data collection was these hospitals, so removing it from the methods cannot be correct. However, I removed the number and wrote the detailed section on data collection in hospitals. Briefly, I also added analysis tools. Lines 33, 38-39.

Conclusion:

social support in the workplace helps young nurses cope better with job stress and protect their mental health and wellbeing. How you conclude that since your study does not investigate social support. Please re-write your conclusion to be consistent with your findings.

Response: I edited it: lines 47-51. 

Keywords: add flexibility word.

Response: I added in keywords. 

Introduction:

The introduction is well-written. However, there are some missing paragraphs for demographic factors included in your study. Why they are important and what the literature says about them. I highly recommend before the method section to write the significant values of your study and make a good connection between your variables.

Response: I added: lines 110-115. 

Method:

-public hospitals: how many?

Response: 10 Grade II public hospitals in Shanghai: line:125. 

-Clarify the data time collection period.

Response: September 20 to December 20, 2023: line: 121. 

-Address the issue of data collection such as how many nurses respond and what is the response rate. (Maybe as an appendix Table).

Response: I added it in results section: lines: 184-187. 

Add a reference for the normal values of CFI, TLI, REMSA…etc in your analysis section.

Response: I added: lines 210-211. 

Results

Appropriate.

Discussion:

-Please highlight your findings instead of repeating your aims.

-being more psychologically flexible. Please how nurses can be??? Write more details in application.

-write more details regarding the connection between your variables.

- I noticed that the majority of references were from China. I respect that. But I suggest strengthening your paper by adding more international studies regarding your topic.

Response: I added some texts in this section. 

Finally, I hope to add clinical implications for nurses to reduce the effect of the night shift on their lives. Some nurses are obligated to work in night shift for several reasons (study, home). Please take this into your mind.

Response: I edited limitation: lines: 326-335. 

References:

Please double-check according to the journal style.

Response: I edited it. 

Tables:

Ensure that abbreviations used in tables are mentioned in the footnotes.

Response: yes, thanks. 

Reviewer #2: 

Thank you for reviewing of this paper. The manuscript is well written in standard English. The statistical Analysis method should be somewhat clear. The recommendation should be revised and rewritten based on your findings.

Response: I edited 

Lines: 326-335.

---

## [Decision Letter · Decision Letter 1]

21 Oct 2024

PONE-D-24-36000R1The effects of psychological flexibility and night shifts on the relationship between mental health and well-being in Chinese nursesPLOS ONE

Dear Dr. Lin,

Thank you for submitting your manuscript to PLOS ONE. After careful consideration, we feel that it has merit but does not fully meet PLOS ONE’s publication criteria as it currently stands. Therefore, we invite you to submit a revised version of the manuscript that addresses the points raised during the review process.

**ACADEMIC EDITOR: **

Dear authors,

After this first round of revisions, Reviewer 1 suggestions have not been fully addressed by the authors.

Please, check again all the comments from the first round of revisions and make appropriate changes.

You should take into account that if these comments are not fully addressed by the authors, the manuscript cannot be accepted for publication and will not be considered again.

I encourage the authors to look carefully to all the comments and make all the necessary modifications in the manuscript without hesitation.

Thank you for understanding.

We look forward to receiving your revised manuscript.

Kind regards,

Javier Fagundo-Rivera, PhD

Academic Editor

PLOS ONE

**Additional Editor Comments:**

Dear authors,

After this first round of revisions, Reviewer 1 suggestions have not been fully addressed by the authors.

Please, check again all the comments from the first round of revisions and make appropriate changes.

You should take into account that if these comments are not fully addressed by the authors, the manuscript cannot be accepted for publication and will not be considered again.

I encourage the authors to look carefully to all the comments and make all the necessary modifications in the manuscript without hesitation.

Thank you for understanding.

Reviewers' comments:

Reviewer's Responses to Questions

**Comments to the Author**

1. If the authors have adequately addressed your comments raised in a previous round of review and you feel that this manuscript is now acceptable for publication, you may indicate that here to bypass the “Comments to the Author” section, enter your conflict of interest statement in the “Confidential to Editor” section, and submit your "Accept" recommendation.

Reviewer #1: (No Response)

Reviewer #2: All comments have been addressed

2. Is the manuscript technically sound, and do the data support the conclusions?

Reviewer #1: Yes

Reviewer #2: Yes

3. Has the statistical analysis been performed appropriately and rigorously? 

Reviewer #1: Yes

Reviewer #2: Yes

4. Have the authors made all data underlying the findings in their manuscript fully available?

Reviewer #1: No

Reviewer #2: Yes

5. Is the manuscript presented in an intelligible fashion and written in standard English?

Reviewer #1: Yes

Reviewer #2: Yes

6. Review Comments to the Author

Reviewer #1: Dear Authors,

**I have noticed that my comments were not taken into consideration**, especially in the Methods and Discussion sections. It is important that these issues are addressed thoroughly to enhance the quality and clarity of your manuscript.

Best regards,

Reviewer #2: Thank you for inviting me to review this paper. The title “The effects of psychological flexibility and night shifts on the relationship between mental health and well-being in Chinese nurses: A systematic review” was very interesting.

My concerns are mentioned below:

Keywords; mental health, nurse, night shift, wellbeing, psychological flexibility make the first letter of each word capitalize.

Overall the paper is well modified

Thank You

7. PLOS authors have the option to publish the peer review history of their article (what does this mean?). If published, this will include your full peer review and any attached files.

Reviewer #1: No

Reviewer #2: **Yes: **Worku Chekol Tassew

---

## [Author Response · Author response to Decision Letter 1]

27 Oct 2024

Reviewer #1: Dear Authors,

I have noticed that my comments were not taken into consideration, especially in the Methods and Discussion sections. It is important that these issues are addressed thoroughly to enhance the quality and clarity of your manuscript.

Best regards. 

Response: Thanks to the first reviewer: I have reviewed your previous comments and revised the text accordingly.

Previous comments of the Reviewer #1: 

Title:

Comment: Please consider revising the title to make no more than 15 words.

Response: I changed the title to: The effects of psychological flexibility and night shifts on mental health and well-being in nurses----- lines: 1-2

Abstract:

Comment: the 10 Grade II public hospitals. I think this information is not necessary. Please add the statistical issues in your abstract.

Response: I edited it; lines 32-40. 

Conclusion:

Comment: social support in the workplace helps young nurses cope better with job stress and protect their mental health and wellbeing. How you conclude that since your study does not investigate social support. Please re-write your conclusion to be consistent with your findings.

Response: I edited it: lines 48-52. 

Comment: Keywords: add flexibility word.

Response: I added in keywords. Line: 53. 

Introduction:

Comment: The introduction is well-written. However, there are some missing paragraphs for demographic factors included in your study. Why they are important and what the literature says about them. I highly recommend before the method section to write the significant values of your study and make a good connection between your variables.

Response: I added some demographic factors in lines 80-88 and 115-119. Also, in lines 125-130, I added the significant values of study.

Method:

Comment: public hospitals: how many? 

Response: 10 Grade II public hospitals in Shanghai were selected.: line:140. 

Comment: Clarify the data time collection period.

Response: September 20 to December 20, 2023: line: 136. 

Comment: Address the issue of data collection such as how many nurses respond and what is the response rate. 

Response: I added it in results section: In general, 612 questionnaires were distributed among nurses. 564 questionnaires were filled by the participants (the response rate was equal to 92.15%). However, only 422 questionnaires were approved (the correct response rate was equal to 74.82%) (27.34 ±4.19 years) and the rest of the questionnaires were not completely filled and therefore were excluded from the study. lines: 201-205. 

Comment: Add a reference for the normal values of CFI, TLI, REMSA…etc in your analysis section.

Response: I added reference in statistical section. GFI >0.95 [37], NFI >0.95 [38], CFI >0.96 [39], AGFI >0.90 [37], RMSEA <0.05 [37], and TLI >0.90 [37] indicated the fit of the model. Lines: 195-196. 

Results

Appropriate.

Discussion:

Comment: being more psychologically flexible. Please how nurses can be??? Write more details in application.

Response: I added some contents about it in discussion section: lines 318-325. (In general, it can be said that psychological flexibility was one of the factors affecting the mental health and well-being of nurses, which was lower in male nurses and younger nurses. Studies show that one of the ways to increase psychological flexibility is the use of Acceptance and Commitment Therapy (ACT) interventions [62, 63]. ACT focuses on pursuing important life areas and goals, like close relationships, fulfilling work, and personal development, even when facing difficult experiences. In ACT, this is achieved by developing psychological flexibility, which helps people stay involved in meaningful activities, even when they have negative thoughts, feelings, or face other challenges [63]). 

Comment: write more details regarding the connection between your variables.

Response: I edited whole of discussion section. 

Comment: I noticed that the majority of references were from China. I respect that. But I suggest strengthening your paper by adding more international studies regarding your topic.

Response: I added some studies from other studies in introduction and discussion sections (Studies from Australia, UK, Jordan, Vietnam, and...). 

Comment: Finally, I hope to add clinical implications for nurses to reduce the effect of the night shift on their lives. Some nurses are obligated to work in night shift for several reasons (study, home). Please take this into your mind.

Response: I added it: lines: 352-362. 

References:

Comment: Please double-check according to the journal style.

Response: I have used Endnote software for writing references. I used the PloS reference writing style. 

Reviewer #2: 

Reviewer #2: Thank you for inviting me to review this paper. The title “The effects of psychological flexibility and night shifts on the relationship between mental health and well-being in Chinese nurses: A systematic review” was very interesting.

My concerns are mentioned below:

Keywords; mental health, nurse, night shift, wellbeing, psychological flexibility make the first letter of each word capitalize.

Overall the paper is well modified

Response: I edited it.

---

## [Decision Letter · Decision Letter 2]

29 Oct 2024

The effects of psychological flexibility and night shifts on mental health and well-being in nurses

PONE-D-24-36000R2

Dear Dr. Lin,

We’re pleased to inform you that your manuscript has been judged scientifically suitable for publication and will be formally accepted for publication once it meets all outstanding technical requirements.

Kind regards,

Javier Fagundo-Rivera, PhD

Academic Editor

PLOS ONE

Additional Editor Comments (optional):

Dear Authors,

thank you for your efforts to assure that your responses are accurate and concise to respond to the Reviewers.

Your manuscript can be accepted.

Congratulations.

Reviewers' comments:

Reviewer's Responses to Questions

**Comments to the Author**

1. If the authors have adequately addressed your comments raised in a previous round of review and you feel that this manuscript is now acceptable for publication, you may indicate that here to bypass the “Comments to the Author” section, enter your conflict of interest statement in the “Confidential to Editor” section, and submit your "Accept" recommendation.

Reviewer #1: All comments have been addressed

2. Is the manuscript technically sound, and do the data support the conclusions?

Reviewer #1: Yes

3. Has the statistical analysis been performed appropriately and rigorously? 

Reviewer #1: Yes

4. Have the authors made all data underlying the findings in their manuscript fully available?

Reviewer #1: (No Response)

5. Is the manuscript presented in an intelligible fashion and written in standard English?

Reviewer #1: Yes

6. Review Comments to the Author

Reviewer #1: Thanks for taking my comments under consideration. I think now that the paper is now ready for publication. Good job and good of luck.

7. PLOS authors have the option to publish the peer review history of their article (what does this mean?). If published, this will include your full peer review and any attached files.

Reviewer #1: No

---

## [Editor Report · Acceptance letter]

5 Nov 2024

PONE-D-24-36000R2 

PLOS ONE

Dear Dr. Lin, 

I'm pleased to inform you that your manuscript has been deemed suitable for publication in PLOS ONE. Congratulations! Your manuscript is now being handed over to our production team.

Kind regards, 

on behalf of

Dr. Javier Fagundo-Rivera 

Academic Editor

PLOS ONE